# Analysis of 16S rRNA Gene Sequence of Nasopharyngeal Exudate Reveals Changes in Key Microbial Communities Associated with Aging

**DOI:** 10.3390/ijms24044127

**Published:** 2023-02-18

**Authors:** Sergio Candel, Sylwia D. Tyrkalska, Fernando Pérez-Sanz, Antonio Moreno-Docón, Ángel Esteban, María L. Cayuela, Victoriano Mulero

**Affiliations:** 1Grupo de Inmunidad, Inflamación y Cáncer, Departamento de Biología Celular e Histología, Facultad de Biología, Universidad de Murcia, 30100 Murcia, Spain; 2Instituto Murciano de Investigación Biosanitaria (IMIB)-Arrixaca, 30120 Murcia, Spain; 3Centro de Investigación Biomédica en Red de Enfermedades Raras (CIBERER), Instituto de Salud Carlos III, 28029 Madrid, Spain; 4Servicio de Microbiología, Hospital Clínico Universitario Virgen de la Arrixaca, 30120 Murcia, Spain; 5Grupo de Telomerasa, Cáncer y Envejecimiento, Servicio de Cirugía, Hospital Clínico Universitario Virgen de la Arrixaca, 30120 Murcia, Spain

**Keywords:** nasopharyngeal microbiome, age differences, sex differences, aging, human microbiome, 16S rRNA sequencing

## Abstract

Functional or compositional perturbations of the microbiome can occur at different sites, of the body and this dysbiosis has been linked to various diseases. Changes in the nasopharyngeal microbiome are associated to patient’s susceptibility to multiple viral infections, supporting the idea that the nasopharynx may be playing an important role in health and disease. Most studies on the nasopharyngeal microbiome have focused on a specific period in the lifespan, such as infancy or the old age, or have other limitations such as low sample size. Therefore, detailed studies analyzing the age- and sex-associated changes in the nasopharyngeal microbiome of healthy people across their whole life are essential to understand the relevance of the nasopharynx in the pathogenesis of multiple diseases, particularly viral infections. One hundred twenty nasopharyngeal samples from healthy subjects of all ages and both sexes were analyzed by 16S rRNA sequencing. Nasopharyngeal bacterial alpha diversity did not vary in any case between age or sex groups. Proteobacteria, Firmicutes, Actinobacteria, and Bacteroidetes were the predominant phyla in all the age groups, with several sex-associated. *Acinetobacter*, *Brevundimonas*, *Dolosigranulum*, *Finegoldia*, *Haemophilus*, *Leptotrichia*, *Moraxella*, *Peptoniphilus*, *Pseudomonas*, *Rothia*, and *Staphylococcus* were the only 11 bacterial genera that presented significant age-associated differences. Other bacterial genera such as *Anaerococcus*, *Burkholderia*, *Campylobacter*, *Delftia*, *Prevotella*, *Neisseria*, *Propionibacterium*, *Streptococcus*, *Ralstonia*, *Sphingomonas*, and *Corynebacterium* appeared in the population with a very high frequency, suggesting that their presence might be biologically relevant. Therefore, in contrast to other anatomical areas such as the gut, bacterial diversity in the nasopharynx of healthy subjects remains stable and resistant to perturbations throughout the whole life and in both sexes. Age-associated abundance changes were observed at phylum, family, and genus levels, as well as several sex-associated changes probably due to the different levels of sex hormones present in both sexes at certain ages. Our results provide a complete and valuable dataset that will be useful for future research aiming for studying the relationship between changes in the nasopharyngeal microbiome and susceptibility to or severity of multiple diseases.

## 1. Background

Among the remaining challenges in biomedical sciences, one of the most important is to fully understand the effect of aging on human biological processes, health, and wellness. In recent years, solid evidence has been collected to support the idea that the microbial communities that inhabit the different anatomical areas of the human body could play a key role in these processes, and there has been much speculation about possible medical interventions [1,2,3,4,5,6]. Although most research has focused on the well-studied gut microbiome, there is growing evidence that variations in microbial communities in other sites of the body are also responsible for wide-ranging health effects [7,8,9,10,11]. The case of the respiratory tract is curious, since the lungs were long believed to be sterile, despite the fact that they are constantly exposed to microorganisms in inhaled air and the upper respiratory tract [12], which has been the main cause that the respiratory microbiome has barely been studied until very recently [13]. However, new culture-independent microbial identification techniques, such as metagenomics, have revealed that the respiratory tract is a dynamic ecosystem, and this has raised the interest of the scientific community in the role of the respiratory microbiota in health and disease [12,13]. 

The human upper respiratory tract that comprises the anterior nares, nasal cavity, sinuses, nasopharynx, Eustachian tube, middle ear cavity, oral cavity, oropharynx, and larynx is the major portal of entry for infectious droplet- or aerosol-transmitted microorganisms [14]. Among these different areas, the nasopharynx is anatomically unique because it presents a common meeting place for the ear, nose, and mouth cavities [15], but has not gained special prominence until the outbreak of the current COVID-19 pandemic [16]. Importantly, dozens of studies have already detected unquestionable correlations between the composition of the nasopharyngeal microbiota and susceptibility to different viral infections in humans [13], and some evidence is emerging, although still controversial, that it may be playing a role in the susceptibility to SARS-CoV-2 infection, too [17]. Elucidating this might shed light on the still unexplained fact that some COVID-19 patients, such as the elderly, are more susceptible and present more severe forms of COVID-19 than others [18].

Large cohort studies of human microbiome data with appropriate controls are particularly valuable, especially of all ages and both sexes, as these datasets are difficult to obtain due to multiple factors, including our long lifespans, heterogeneity in consent and other sample access issues, and because of socioeconomic confounds. Therefore, human studies have tended to focus on a specific component of the lifespan, such as infanthood, or studies of the elderly, rather than examining variation across an entire population. Knowledge about the relationships between changes in the nasopharyngeal microbiota and susceptibility to viral infections is a good example of this, since most studies have focused only on children [13]. 

A crucial factor to consider in studies of variation across the lifespan is sex as a biological variable. For aging research, this includes the understanding that females and males may have different aging trajectories [19,20,21], including in key systems such as the digestive tract. For example, the gut microbiome and sex hormones may interact to predispose women to autoimmune diseases [22] and dietary interventions are known to have sex-specific effects on gut microbiota [23]. There are no studies analyzing the possible sex-associated differences in the nasopharyngeal microbiome at different life stages.

Here, we analyze, for the first time, the diversity and relative abundance of the nasopharyngeal microbiota across the whole lifespan in 120 healthy individuals of all ages and both sexes, the taxonomic changes in the nasopharynx associated to age or sex, and the possible biological relevance of several taxa whose frequency of appearance in the population is high. We therefore provide a very comprehensive and valuable dataset that will be the base for future research aimed at identifying relationships between age- and sex-associated changes in nasopharyngeal microbiome and susceptibility to or severity of the diseases of interest.

## 2. Results

### 2.1. Data Annotation and Sample Overview

A total of 120 nasopharyngeal microbiomes from 120 healthy individuals were analyzed. A total of 4,538,196 high-quality 16S rRNA sequences ranging from 10,627 to 256,449 sequences per sample (mean = 37,818.3; median = 33,169) were obtained after quality control analyses and OTU filtering. The 16S rRNA sequences were binned into 128 families, 250 genera and 561 species. The most abundant families were Staphylococcaceae (12.14%), Burkholderiaceae (11.52%), Carnobacteriaceae (11.48%) and Corynebacteriaceae (9.47%). The most abundant genera were *Staphylococcus* (13.06%), *Dolosigranulum* (11.99%), *Corynebacterium* (10.18%) and *Ralstonia* (10.08%). The most abundant species were *Dolosigranulum pigrum* (24.55%), *Ralstonia pickettii* (19.02%), *Corynebacterium pseudodiphtheriticum* (4.87%) and *Propionibacterium acnes* (4.85%). We excluded one sample with an abnormally high proportion of *Chlamydophila*, which is an indication of an abnormal sampling or pathological disorder of this individual ‘C_A1_M8’. To reveal age-related progression of nasopharyngeal microbiota, we divided the samples into six age groups, each divided into females and males to be able to also study possible sex-associated differences (Appendix A). There were 20 samples in each age group and 10 samples in each sex group within them, except for the first age group (A1: 1–20 years) where one male had to be excluded as indicated above (Appendix A).

We sought to determine the ways in which different samples were grouped according to their OTU composition. To that end, we applied nonmetric multidimensional scaling (NMDS), which is a powerful statistical tool that enables complex multivariate data sets to be visualized in a reduced number of dimensions, to determine the clustering patterns of samples according to their Bray–Curtis distances (which were calculated based on the relative abundance matrix of the 250 genera across the 119 samples) (Figure 1). The analysis of similarities (ANOSIM), which is a non-parametric statistical test, was used to analyze whether there were statistically significant differences among the different age groups included in this study. Thus, even though the samples apparently did not form distinct clusters when viewed using this approach as they appeared mostly intermixed and the different confidence ellipses overlapped each other, the differences between the age groups A1–A4 (ANOSIM statistic, 0.1075; significance, 0.016) and A1–A5 (ANOSIM statistic, 0.1075; significance, 0.016) were significant according to ANOSIM (Figure 1). Similar result was obtained when focusing on possible differences between the two sexes, as samples from females and males also appeared completely intermixed and did not form any groups, and significant differences were not detected according to ANOSIM (Figure 1).

### 2.2. Bacterial Diversity in the Nasopharynx of Healthy Individuals Is Stable throughout Lifespan

The fact that significant changes in bacterial diversity throughout life had previously been described in the well-studied gut microbiota of healthy individuals [24] prompted us to test whether similar changes occur in the nasopharynx by analyzing the alpha diversity, referred to as within-community diversity [25], for the different age and sex groups established for this study (Appendix A). However, the Shannon’s diversity index, which measures evenness and richness of communities within a sample, did not show any statistically significant changes in bacterial diversity among the different age groups (Figure 2a). Moreover, alpha diversity also did not vary as a function of sex when the Shannon index was calculated considering all individuals of all ages included in this study (Figure 2b), nor when the same analysis was performed comparing females and males within each age group (Figure 2c). The use of other indexes commonly used to measure alpha diversity, such as the inverse Simpson’s diversity index, which is an indication of the richness in a community with uniform evenness that would have the same level of diversity (Appendix A), or the Chao1 index, which measures the total richness of communities within a sample (Appendix A), confirmed the absence of any statistically significant differences in bacterial diversity between the different age groups (Appendix A) or between females and males (Appendix A). Therefore, all these results together suggest that contrary to what occurs in other anatomical areas, such as the gut where bacterial diversity decreases with aging [24], it remains stable in the nasopharynx of healthy people over time, without notable changes at any stage of life, not even in very young people or in the elderly over 70 years of age (Figure 2a and Appendix A). Curiously, another interesting finding provided by this work, for the first time, is that there are no significant differences when comparing bacterial diversity in the nasopharynx of healthy females and males (Figure 2b,c and Appendix A), regardless of the stage of life studied and the important hormonal differences that exist between both sexes at certain ages. 

### 2.3. Age- and Sex-Associated Changes in Relative Abundance of Bacterial Taxa in the Nasopharynx of Healthy Individuals

To determine differences in nasopharyngeal taxa abundance among age and sex classes, we compared the nasopharyngeal microbiome of healthy women and men in the six different age groups established for this study (Appendix A). The first general analysis at the phylum level revealed that Firmicutes and Proteobacteria relative abundances showed opposite kinetics with aging (Figure 3a–f), while the relative abundance of Firmicutes, which is the majority phylum in the youngest individuals (50%) (Figure 3a), clearly decreased with aging reaching its lowest values in subjects in their 50s and 60s (21% and 27%, respectively) (Figure 3d,e). The relative abundance of Proteobacteria presented its lowest value in the youngest people (24%) (Figure 3a) and increased in older individuals, peaking in subjects who were in their 50s (53%) (Figure 3d). The relative abundance of other phyla, such as Actinobacteria, Bacteroidetes, Tenericutes, or Cyanobateria, remained more stable throughout life (Figure 3a–f). Moreover, when we continued working at the phylum level and searched for any abundance differences between the nasopharyngeal microbiota of females and males, we determined that, interestingly, the results were almost identical for both sexes within the two age groups containing the youngest (1–20 years) (Appendix A) and oldest (>70 years) (Appendix A) individuals. However, several differences between females and males were observed in other age groups, notably a higher relative abundance of Actinobacteria in males in their 20s and 30s compared to females of the same age group (20% vs. 7%) (Appendix A), a higher relative abundance of Proteobacteria in males in their 40s compared to females of the same age group (46% vs. 29%) (Appendix A), a higher relative abundance of Proteobacteria in females in their 50s compared to males of the same age group (57% vs. 47%) (Appendix A), and a higher relative abundance of Firmicutes and lower of Actinobacteria in females in their 60s compared to males of the same age group (37% vs. 20% and 10% vs. 22%, respectively) (Appendix A). Therefore, the fact that there were differences in taxa abundance when comparing the nasopharyngeal microbiota of females and males in most of the age groups studied (individuals between 21 and 70 years of age) (Appendix A), but not within the two age groups containing the youngest and oldest individuals (people between 1 and 20 and older than 70 years of age) (Appendix A), suggests that the different levels of sex hormones present in both sexes at different life stages might be modulating the nasopharyngeal microbiome. 

Next, we proceeded further in this study and moved to the family level, finding that 24 distinct bacterial families presented an average abundance of >1% in at least one of the age groups studied (Figure 3g). Our analyses revealed the dominant family in each of the age groups: Staphylococcaceae in A1, Carnobacteriaceae in A2, Staphylococcaceae in A3, Burkholderiaceae in A4, Streptococcaceae in A5, and Staphylococcaceae in A6 (Figure 3g). Thus, it is curious that although Proteobacteria was the majority taxa in all the age groups when analyzing the nasopharyngeal microbiota at the phylum level (Figure 3a–f), the dominant family in all the age groups belongs to the Firmicutes phylum, except in the case of age group A4 where the family Burkholderiaceae, which belongs to the Proteobacteria phylum, was the most abundant taxa at family level (Figure 3g). Differences in the relative abundance of some families were detected when compared between age groups, but without following any easily interpretable pattern (Figure 3g). Analysis of differential taxa abundance at the family level between age groups, but separately in females and males, did not show any relevant sex-associated differences between sexes regarding the dominant bacterial families in each age group compared to the results described above for both sexes together (Figure 3g), excepting that Burkholderiaceae was the most abundant family in males of age group A3 instead of Staphylococcaceae, and that Corynebacteriaceae is the dominant family in males of age group A5 instead of Streptococcaceae (Appendix A). As mentioned above when working with females and males together, differences in relative abundance were detected in some families when comparing between different age groups in both sexes, but without following any easily interpretable patterns (Appendix A). Visualization of taxa abundance at the family level in all the individuals included in this study showed that most of them had a very diverse microbiome, with a high number of families with relative abundance of >1% (Appendix A). However, a few individuals had one dominant family that represented the majority of their nasopharyngeal microbiomes; these families tended to be Burkholderiaceae, Carnobacteriaceae and Staphylococcaceae (Appendix A).

Working at the genus level, we determined that 24 bacterial genera presented an average abundance of >1% in at least one of the age groups studied (Figure 3h). Moreover, our results showed that *Staphylococcus* was the dominant genus in age group A1, *Dolosigranulum* in A2, *Staphylococcus* in A3 and A6, *Ralstonia* in A4, and *Streptococcus* in A5 (Figure 3h). No relevant differences were detected when comparing taxa abundance in the nasopharynx of females and males at the genus level within the age groups studied (Appendix A). Similar to what was described above at the family level, visualization of taxa abundance at the genus level in all the individuals included in this study showed that most of them had a high number of genera with relative abundance of >1%, with a few individuals presenting one dominant genus (mostly *Dolosigranulum* or *Staphylococcus*) that represented the majority of their nasopharyngeal microbiomes (Appendix A). Next, we focused on those genera whose relative abundance were significantly different between the different age groups established in this study, as this could help us to identify changes in the nasopharyngeal microbiota that are characteristic of aging. Thus, our analyses revealed that there were statistically significant differences (adjusted *p*-value < 0.05) in relative abundance between the distinct age groups in 11 bacterial genera: *Acinetobacter*, *Brevundimonas*, *Dolosigranulum*, *Finegoldia*, *Haemophilus*, *Leptotrichia*, *Moraxella*, *Peptoniphilus*, *Pseudomonas*, *Rothia* and *Staphylococcus* (Figure 4a and Appendix A). Interestingly, most of these statistically significant differences in relative abundance between the age groups for the 11 mentioned genera were between age groups A1 or A6, which include individuals between 1 and 20 years of age and over 70 years old, respectively, and the rest of age groups (18 out of 37 cases for the age group A1 and 16 out of 37 cases for the age group A6) (Appendix A). Among these statistically significant changes detected, it should be noted that *Acinetobacter* was the only genus whose relative abundance in the nasopharynx clearly increased progressively throughout life, peaking in individuals older than 70 years of age (Figure 4a and Appendix A). In the cases of *Dolosigranulum* and *Rothia*, their relative abundance drastically increased and decreased, respectively, in individuals over 70 years of age, compared to middle-aged people in their 50s and 60s (Figure 4a and Appendix A). Changes among the different age groups of *Finegoldia*, *Leptotrichia* and *Haemophilus* were also interesting, as their relative abundance was markedly reduced in elderly people over 70 years of age, even though they were present at other ages throughout life, mainly during middle age (Figure 4a and Appendix A). The case of *Haemophilus* was particularly intriguing, as while its relative abundance was at least 10% of the nasopharyngeal microbiota composition in age groups A1-A5 (if we consider only these 11 genera that present statistically significant differences between age groups), it dramatically decreased in the group of individuals over 70 years old (Figure 4a and Appendix A). Furthermore, it is noteworthy that the bacterial genera *Brevundimonas*, *Finegoldia*, *Leptotrichia* and *Peptoniphilus* presented a very low relative abundance in the youngest individuals, who are between 1 and 20 years old, compared to other age groups (Figure 4a and Appendix A). Finally, although *Moraxella* and *Staphylococcus* showed significant differences in relative abundance between the distinct age groups in several cases, these differences did not seem to follow any easily interpretable pattern relating relative abundance levels to a particular life stage (Figure 4a and Appendix A). Next, we wondered whether the significant differences in relative abundance between age groups observed for these 11 bacterial genera were due to sex-associated differences. Visualization of taxa abundance in females and males separately, considering only these 11 genera, showed no notable differences between both sexes (Appendix A). The only exception was a higher relative abundance of *Dolosigranulum* in males in their 20s and 30s compared to females of the same age, because this genus was clearly dominant in five males from that age group while only in one female of the same age (Appendix A). Besides this observation regarding *Dolosigranulum*, analyzing the taxa abundance in all the individuals included in this study also revealed that in most people, 1 out of these 11 genera was dominant compared to the relative abundance of the other 10 genera (Appendix A). Interestingly, 8 out of the 11 genera, excepting *Bevundimonas*, *Finegoldia* and *Peptoniphilus*, were determined to be the dominant genus in at least one individual, demonstrating that nasopharyngeal taxonomic composition at this level can be very variable between different individuals, even if they belong to the same age or sex groups (Appendix A).

### 2.4. Identification of Potentially Biologically Relevant Bacterial Genera by Analyzing Their Frequency of Appearance in the Nasopharynx of Healthy Individuals

After analyzing taxa abundance at the phylum, family, and genus levels, looking for differences between age and sex groups, we decided to apply another strategy to attempt to identify bacterial genera whose presence in the nasopharynx might be characteristic of a certain life stage, independently on their abundance levels. Thus, based on the idea that in some cases the presence of a genus within a certain sex or age group but not in others can be biologically relevant, even if its abundance is low, we studied the frequency with which each genus appeared in the individuals included in this study by visualizing, for each genus, the percentage of people from each age group in which it is present. Firstly, we analyzed this in the 11 genera that presented significant differences between the distinct age groups, with the aim of checking whether these genera were present in a high percentage of individuals and, therefore, that the relative abundance results previously shown in Figure 4a were reliable. Indeed, our results showed that these 11 genera appeared with a high frequency in the individuals included in this study, especially in the cases of *Acinetobacter*, *Dolosigranulum*, *Haemophilus*, *Moraxella*, *Pseudomonas* and *Staphylococcus* (Figure 4b). These data also revealed that several genera, such as *Brevundimonas*, *Finegoldia* and *Peptoniphilus*, appeared less frequently in the youngest people than in the other age groups, while the frequency of appearance of others, such as *Rothia*, decreases in the oldest people (Figure 4b). Interestingly, it coincides that these genera that appear most frequently in the individuals included in this study are also the ones that appear most often as the dominant genus in the taxonomic composition of individuals (Appendix A). Next, we identified several bacterial genera that could be biologically relevant as part of the nasopharyngeal microbiota, although they did not show any significant differences in relative abundance between age or sex groups. This was (i) because their frequency of appearance in the nasopharynx of the healthy population was very high, as in the case of *Anaerococcus*, *Burkholderia*, *Campylobacter*, *Delftia*, *Prevotella*, *Neisseria*, *Propionibacterium*, *Streptococcus*, *Ralstonia*, *Sphingomonas* and *Corynebacterium* (Figure 5a); (ii) because their frequency of appearance drastically increased with age, such as the cases of *Faecalibacterium*, *Stenotrophomonas* and *Phascolarctobacterium* (Figure 5b); or (iii) because their frequency of appearance drastically decreased with age, such as the cases of *Aggregatibacter*, *Gemella* and *Fusobacterium* (Figure 5c).

## 3. Discussion

Comparing bacterial species richness in the nasopharynx of healthy individuals in the different age groups established for this study and between females and males revealed no age- or sex-associated significant differences in alpha diversity. These results, which were confirmed by using three of the most reliable and commonly used alpha diversity indexes, such as the Shannon’s diversity index, the inverse Simpson’s diversity index, and the Chao1 diversity index, indicate that the nasopharyngeal microbiome is highly stable and robust to perturbations throughout life as well as between sexes within each of the different age groups in healthy subjects. Alpha diversity could be expected to be lower in the youngest individuals compared to older individuals in all the anatomical areas, as the fetus is sterile until the moment of birth, when the newborn begins to be progressively colonized by microorganisms until its definitive microbiota is established [26]. This is exactly what happens, for example, with the gut microbiome of infants, whose alpha diversity is consistently determined to be lower than in adults [27,28], probably due to the introduction of new diversity from food, which increases with the consumption of foods other than breast milk [29]. Detailed metagenomic studies, which should include samples collected at multiple time points during childhood and during adulthood, would be needed to determine whether something similar occurs in the nasopharynx. In the case of the present study, we would not be able to detect such differences in diversity between the youngest individuals and the rest of age groups, if they exist, for two reasons: (i) because our first age group, which includes the youngest individuals, is very broad and encompasses individuals up to 20 years old, so any differences from young children could be diluted; (ii) because as we were not interested specifically in children, but in broader age groups spanning all stages of life, we excluded children younger than 1 year of age from this study, as their highly changing microbiota in formation could bias the analyses we were interested in. Interestingly, although it is known that there are relevant sex-associated differences in diversity in other anatomical areas such as the gut, and that such differences are probably due to the different levels of sex hormones between both sexes [30], we did not observe any significant diversity differences between females and males, even in the age groups where sex hormones levels should be quite different. It is also curious that, contrary to what occurs in the gut where the alpha diversity of the microbiota decreases with aging [24], we did not observe a similar reduction in diversity in the nasopharynx of older individuals. This difference between both anatomical areas might be explained by the fact that decreased microbial diversity of the gut in older subjects is associated with chronological age, number of concomitant diseases, number of medications used, increasing coliform numbers, and changes in diet [31], and although chronological age affects both anatomical areas in a similar way, it seems probable that other of these factors might affect the gut microbiota in a more intense way compared to the nasopharyngeal microbiota. In summary, we can say that the diversity of the nasopharyngeal community is very stable throughout life and between sexes.

Our analyses of taxa abundance in the nasopharynx at the phylum level revealed the existence of sex-associated differences within the age groups including individuals between 21 and 70 years of age, but not in the youngest and oldest people. As previously mentioned for the alpha diversity results, multiple sex-associated differences in taxonomic composition have also been described for the human gut microbiome [30], and numerous studies have reported evidence to support the idea that levels of sex hormones, such as progesterone [32], androgen [33], and estrogen [34], regulate its composition [35,36]. The effects of sex hormones on other microbial niches, such as the human vaginal microbiota, have also recently been demonstrated [37]. Although it has been shown that estrogen stimulation (hormone/gender effect) in the upper respiratory tract mucosa could reduce virus virulence by improving both nasal clearance and local immune response [38], the relationship between sex hormone levels and the nasopharyngeal microbiota has so far not been directly observed in clinical studies. However, the fact that our results reveal differences between females and males at the phylum level in all the age groups, except for those where differences in sex hormones levels do not exist or are not so strong (>70 years and 1–20 years old), suggests that sex hormones might be modulating the taxonomic composition of the healthy human nasopharynx. Note that although differences in sex hormones could be expected to be relevant in the age group that includes individuals between 1 and 20 years of age since pubescents and adolescents are part of this group, we assumed that such differences should not be so significant in our case because 70% of individuals in this age group are prepubescent in our study.

We might be tempted to think that the microbiota of nearby anatomical sites that are closely related in terms of structure and function should be practically identical. However, the reality seems to be much more complex. A good example of this is that although nasopharynx and nose are adjacent, and previous metagenomic studies comparing the microbiome of both anatomical areas have revealed a clear continuity, there are important differences between the two sites and even niche-specific bacteria [39]. Furthermore, this study also reported an evident heterogeneity among participants, since the nasopharyngeal microbiome of half of them was dominated by *Moraxella*, *Streptococcus*, *Fusobacterium*, *Neisseria*, *Alloprevotella* or *Haemophilus*, while in the other half it contained an intermixed bacterial profile where *Staphylococcus*, *Corynebacterium*, and *Dolosigranulum* seemed to be important bacterial members with varying relative abundances [39]. Our taxa abundance analyses at the genus level only detected statistically significant relative abundance differences between the different age groups for 11 bacterial genera: *Acinetobacter*, *Brevundimonas*, *Dolosigranulum*, *Finegoldia*, *Haemophilus*, *Leptotrichia*, *Moraxella*, *Peptoniphilus*, *Pseudomonas*, *Rothia,* and *Staphylococcus*. Interestingly, most of the 37 statistically significant differences detected between the different age groups for these 11 genera appear when comparing age groups A1 and A6 with the rest of the age groups (18 out of 37 and 16 out of 37, respectively). Therefore, these results reveal that, in terms of relative abundance of bacterial genera, the nasopharyngeal microbiota of the youngest and oldest subjects is more different from that of the other age groups than that of any other age group. Among these age-associated changes, from a clinical perspective, it is particularly concerning that *Dolosigranulum*, which is an opportunistic pathogen that causes pneumonia in elderly patients [40], is overrepresented in the nasopharynx of individuals over 70 years of age compared to middle-aged subjects. This suggests that the relative abundance of *Dolosigranulum* may be higher in elderly people due to the process of immunosenescence that occurs in them [41,42], or that its higher abundance may be due to other unidentified age-related factors. Nevertheless, the relevance of *Dolosigranulum* in the nasopharynx deserved further investigation, since nasal administration *Dolosigranulum pigrum* 040417 to mice increased the resistance against respiratory syncytial virus (RSV) and *Streptococcus pneumoniae* [43,44]. However, other strains of the same species failed to protect mice against these pathogens [43,44]. Another interesting observation of our study is that *Haemophilus* that causes pneumonia mainly in elderly people [45] is underrepresented precisely in individuals over 70 years of age. This suggests that their lower relative abundance in elderly subjects is due to other unidentified age-related factors, and that the elderly are much more susceptible to opportunistic infections caused by this bacterium, probably due to the previously mentioned process of immunosenescence. Further research will be necessary to elucidate the precise reason for this. Something similar could be said for *Rothia*, as its relative abundance also decreases drastically in people over 70 years of age while it is known to cause pneumonia mostly in aged individuals [46]. It is worth noting that, regardless of their relative abundance or whether they show statistically significant differences between age or sex groups, those bacterial genera that are present in most individuals or whose frequency of appearance changes drastically throughout life could be relevant from a biomedical and ecological point of view. Based on this idea, we highlight *Anaerococcus*, *Burkholderia*, *Campylobacter*, *Delftia*, *Prevotella*, *Neisseria*, *Propionibacterium*, *Streptococcus*, *Ralstonia*, *Sphingomonas* and *Corynebacterium* as candidate bacterial genera that could be playing an important role as they are present in the nasopharynx of most healthy individuals. In addition, we propose *Faecalibacterium*, *Stenotrophomonas* and *Phascolarctobacterium* as candidate bacterial genera that could be playing a relevant role, as their frequency of appearance in the nasopharynx of healthy subjects increases progressively throughout life, and *Aggregatibacter*, *Gemella* and *Fusobacterium* because their frequency of appearance in the nasopharynx decreases drastically and progressively as healthy people age. Elucidating the biomedical relevance of all these bacterial genera which are part of the healthy nasopharyngeal microbiota and determining their potential involvement in health and disease at different stages of life is certainly an exciting topic for future work.

Our study has several limitations. This was an observational, retrospective study, and collection of data was not standardized in advance. The 16S rRNA gene sequencing approach to study the microbiota could introduce bias in the obtained data because this method does not allow the study of the whole microbiome, but only the genera amplified by PCR. The taxonomic assignment at the species level may not be fully accurate. Nevertheless, it is the most common technique to study microbiota in clinical samples. Moreover, it was not possible to obtain serial samples. Furthermore, the groups are small, particularly the sex groups within each age group, so the study may have been underpowered to detect certain associations. Finally, we could not access any sociodemographic, environmental, lifestyle, or medical information of subjects enrolled in this study, which would have been helpful to better understand the characteristics of the cohort.

Although multiple studies have analyzed the microorganisms present in the nasopharynx in different contexts before this work, the characteristics of the healthy and mature human nasopharyngeal microbiota was largely unknown since (i) most studies focused on children or elderly people, (ii) confounding factors such as external drivers that alter it are not well known to date, and (iii) focus is generally shifted to its variation in diseases. With this work, we fill this important gap in knowledge. However, further research will be necessary to elucidate the effects of the nasopharyngeal taxonomic composition as well as the age- and sex-associated changes described here on the susceptibility of certain individuals to infectious diseases. Studying the case of the elderly people in detail will be particularly interesting from a biomedical and clinical perspective, since their nasopharyngeal microbiota is significantly different from that of younger subjects, and they are known to be much more susceptible to multiple infectious diseases, most notably COVID-19 [47]. Therefore, we hypothesize that there may be some correlation between the taxonomic composition in the nasopharynx of the elderly and their increased susceptibility to COVID-19, but this will be a challenge for future metagenomic studies that should include different age groups, both sexes, and patients infected with SARS-CoV-2 who have developed the disease with different severity.

## 4. Methods and Materials

### 4.1. Sample Selection, Collection, and Classification

Due to the available economic resources, we randomly selected 120 nasopharyngeal samples from a cohort of 6354 healthy subjects belonging to the Health Area I of the Region of Murcia (Spain) who voluntarily provided their samples between 27 August 2020 and 8 September 2020 for diagnostic purposes and tested negative for SARS-CoV-2 infection. Nasopharyngeal swabs were obtained by approaching the nasopharynx transnasally and stored in Universal Transport Medium (UTM): Viral Transport medium (COPAN Diagnostics Inc., Murrieta, CA, USA). Nucleic acid extraction was performed using the automatized system Nuclisens EasymaG (bioMérieux, Madrid, Spain) based on the ability of silica to bind DNA and RNA in high salt concentrations (Boom technology). The polymerase chain reaction (PCR) kit used to verify that all the samples were negative for SARS-CoV-2 infection was Novel Coronavirus (2019-nCoV) Real Time Multiplex RT-PCR kit (Detection for 3 Genes), manufactured by Shanghai ZJ Bio-Tech Co., Ltd. (Liferiver Biotech, la Jolla, CA, USA) and the CFX96 Touch Real-Time PCR Detection System (BioRad, Madrid, Spain).

To facilitate the study of age- and sex-associated changes in the nasopharyngeal microbiota throughout life, and to ensure that the sample size of all the age and sex groups were homogeneous, we decided on an experimental design that distributed the 120 nasopharyngeal samples that we could analyse into six age groups with 20 individuals each, of which 10 were females and the other 10 were males (Appendix A). For this, the 6354 healthy subjects of our parent cohort were divided into their age matched groups and numbered, and then randomly obtained numbers were used to select 10 females and 10 males from each of the age groups. Random numbers were generated in RANDOM.ORG, which is a True Random Number Generator (TRNG) that generates true randomness via atmospheric noise, unlike the most common and less trustworthy Pseudo-Random Number Generators (PRNGs) [RANDOM.ORG: True Random Number Service. Available at: https://www.random.org]. According to the exclusion criteria we established for this study, (1) individuals younger than 1 year of age were disqualified because the microbiome of infants is known to be highly fluctuating with age, and (2) subjects who were tested for SARS-CoV-2 infection because they had respiratory or any other kind of symptoms were also excluded to avoid the enrolment of individuals who could have any infection or disease that could alter their nasopharyngeal microbiota although they were not infected by SARS-CoV-2.

### 4.2. Amplification, Library Preparation, and Sequencing

Bacterial identification was performed by sequencing the 16S rRNA gene’s hypervariable regions. The 16S rRNA gene was amplified by multiplex PCR using Ion Torrent 16S Metagenomics kit (Ion Torrent, Thermo Fisher Scientific Inc., Alcobendas, Spain), with two sets of primers, which targets regions V2, V4, and V8, and V3, V6–7, and V9, respectively. Amplification was carried out in a SimpliAmp thermal cycler (Thermo Fisher Scientific Inc., Alcobendas, Spain) running the following program: denaturation at 95 °C for 10 min, followed by a 3-step cyclic stage consisting of 25 cycles of denaturation at 95 °C for 30 s, annealing at 58 °C for 30 s, and extension at 72 °C for 20 s; at the end of this stage, the program concludes with an additional extension period at 72 °C for 7 min and the reaction is stopped by cooling at 4 °C. The resulting amplicons were tested by electrophoresis through 2% agarose gels in tris-acetate-EDTA (TAE) buffer, purified with AMPure^®^ XP Beads (Beckman Coulter, Inc, Atlanta, GA, USA), and quantified using QubitTM dsDNA HS Assay Kit in a Qubit 3 fluorometer (Thermo Fisher Scientific Inc., Alcobendas, Spain). 

A library was generated from each sample using the Ion Plus Fragment Library Kit (Ion Torrent), whereby each library is indexed by ligating Ion Xpress™ Barcode Adapters (Ion Torrent) to the amplicons. Libraries were purified with AMPure^®^ XP Beads and quantified using the Ion Universal Library Quantitation Kit (Ion Torrent, Thermo Fisher Scientific Inc., Alcobendas, Spain) in a QuantStudio 5 Real-Time PCR Instrument (Thermo Fisher Scientific Inc., Alcobendas, Spain). The libraries were then pooled and clonally amplified onto Ion Sphere Particles (ISPs) by emulsion PCR in an Ion OneTouch™ 2 System (Ion Torrent) according to the manufacturer´s instructions. Sequencing of the amplicon libraries was carried on an Ion 530™ Kit (Ion Torrent) on an Ion S5™ System (Ion Torrent). After sequencing, the individual sequence reads were filtered by the Torrent Suite™ Software v5.12.1 to remove low quality and polyclonal sequences.

### 4.3. Bioinformatics and Statistical Analysis

The obtained sequences were analyzed and annotated with the Ion Reporter 5.18.2.0 software (Thermo Fisher Scientific Inc., Alcobendas, Spain) using the 16S rRNA Profiling workflow 5.18. Clustering into OTUs and taxonomic assignment were performed based on the Basic Local Alignment Search Tool (BLAST) using two reference libraries, MicroSEQ^®^ 16S Reference Library v2013.1 and the Greengenes v13.5 database. For an OTU to be accepted as valid, at least ten reads with an alignment coverage of ≥90% between hit and query were required. Identifications were accepted at the genus and species level with sequence identity of >97% and >99%, respectively. Annotated OTUs were then exported for analysis with R (v.4.1.2) (https://www.R-project.org/), where data were converted to phyloseq object [48] and abundance bar plots were generated. Data were converted to DESeq2 object [49] that uses a generalized linear model based on a negative binomial distribution to calculate differential abundance between groups. Thus, the differential abundance analysis was conducted according to the phyloseq package vignette with bioconductor DESeq2 (https://bioconductor.org/packages/devel/bioc/vignettes/phyloseq/inst/doc/phyloseq-mixture-models.html#import-data-with-phyloseq-convert-to-deseq2, accessed on 10 January 2022). The raw abundance matrix was imported into phyloseq object (as specified in the documentation of phyloseq with DESeq2) and subsequently converted to DESeq2 object. Then, estimated size factors were used with the DESeq2 function to obtain the differential abundance. DESeq automatically searches for outliers and, if possible, replaces the outlier values estimating mean-dispersion relationship. If it is not possible to replace, *p*-values are replaced by NA. R (v.4.1.2) was also used to perform a non-metric multidimensional scaling (NMDS) analysis on Bray–Curtis dissimilarity measures among samples based on relative OTU abundances (i.e., percentages). The relative abundances of OTUs were also used to test for statistically significant differences among age and sex groups. Group OTU compositions were compared through the non-parametric statistical tool ANOSIM. The 90% confidence data ellipses for each of the age groups were plotted. Alpha diversity was estimated based on Chao1, Shannon, and Inverse-Simpson indices by using the phyloseq package. To test for statistically significant differences between pairwise groups in alpha diversity, the non-parametric Wilcoxon test was used. Frequency of appearance was obtained by calculating the percentage of individuals in each age group in which that taxon occurs. The bar plots aggregated by groups (age and/or gender) show the aggregated relative abundance (sum of relative abundances). Krona charts that aid in the estimation of relative abundances even within complex metagenomic classifications were generated as previously described [50]. All the other graphs were generated with the R package ggplot2 version 3.3.3., including the confidence data ellipses which were plotted using the ‘stat_ellipse’ function, also from this package [51].

## 5. Conclusions

Our study shows that bacterial diversity in the nasopharynx of healthy subjects remains very stable and resistant to perturbations throughout the whole life and in both sexes. Age-associated changes in taxa abundance were observed at phylum, family, and genus levels, as well as several sex-associated changes probably due to the different levels of sex hormones present in both sexes at certain ages. We provide a complete and valuable dataset that will be useful for future research aiming for studying the relationship between changes in the nasopharyngeal microbiome and susceptibility to or severity of multiple diseases.

## Figures and Tables

**Figure 1 ijms-24-04127-f001:**
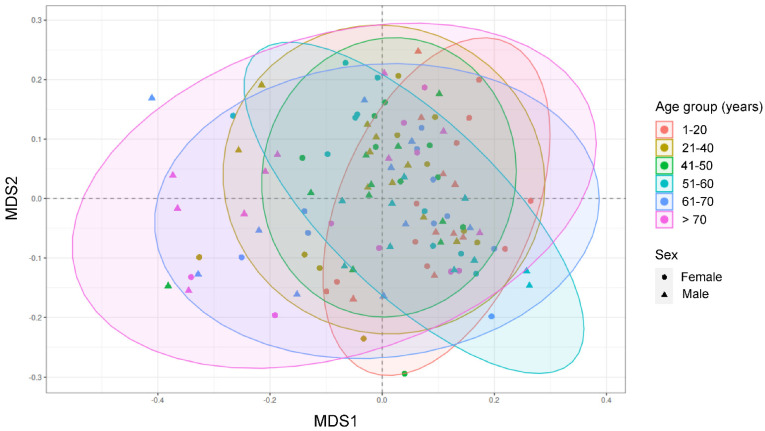
Microbial community composition. Nonmetric multidimensional scaling (NMDS) plot of the Bray–Curtis distances which were calculated using the relative abundance of the 250 genera across the 119 samples as input. Each sample is represented by one dot, colored according to age, and shaped according to sex. The 90% confidence data ellipses are shown for each age group.

**Figure 2 ijms-24-04127-f002:**
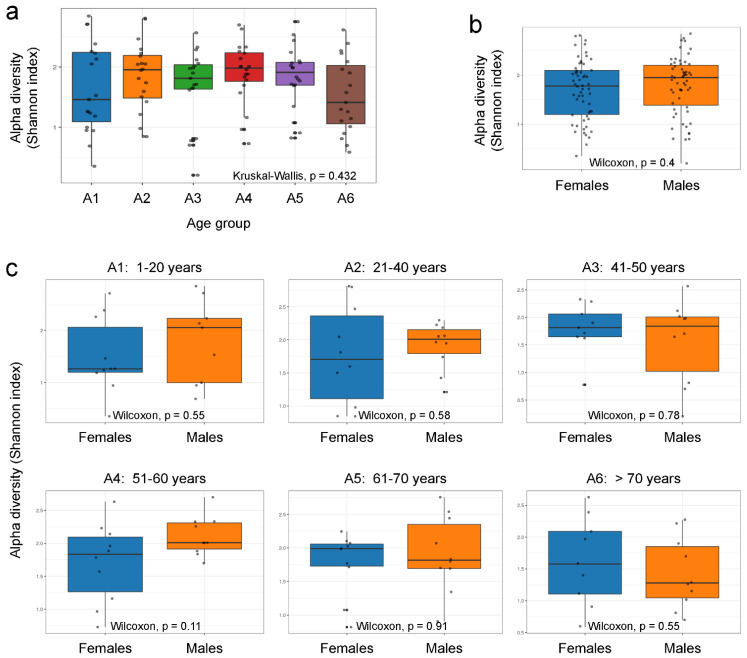
Comparison of alpha diversity parameters across the age and sex groups studied. Box-whisker plots of the alpha diversity Shannon index and its comparison using the Kruskal–Wallis test among the different age groups established for this study (**a**), and the Wilcoxon signed-rank test between females and males (**b**,**c**). Each sample is represented by one dot. The age group A1 includes subjects between 1 and 20 years old, A2 between 21 and 40, A3 between 41 and 50, A4 between 51 and 60, A5 between 61 and 70, and A6 includes individuals over 70 years of age (Appendix A).

**Figure 3 ijms-24-04127-f003:**
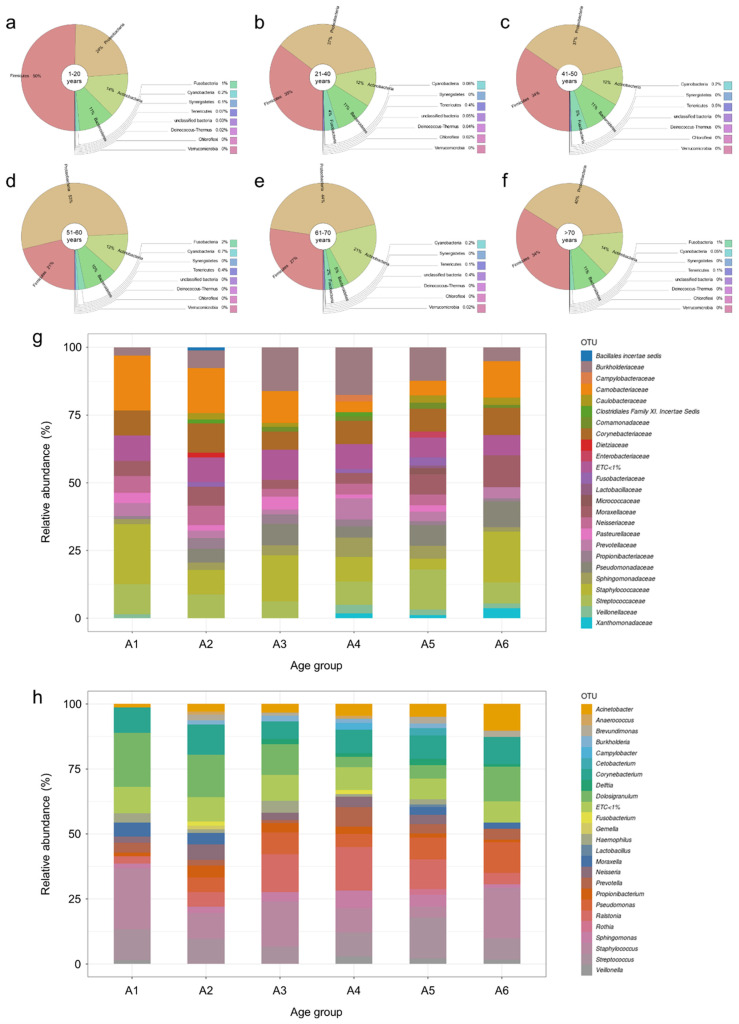
Taxonomic composition and age-associated metagenomic changes in the nasopharynx of healthy individuals. (**a**–**f**) Krona charts showing the bacterial community composition at the phylum level in the indicated age groups. Stacked bar charts showing the relative abundance (%) of bacterial phyla. (**g**) Stacked bar charts showing the relative abundance (%) of bacterial families in the indicated age groups. (**h**) Stacked bar charts showing the relative abundance (%) of bacterial genera in the indicated age groups. For clarity, only bacterial families (**g**) and genera (**h**) with average abundance >1% at each age group are shown.

**Figure 4 ijms-24-04127-f004:**
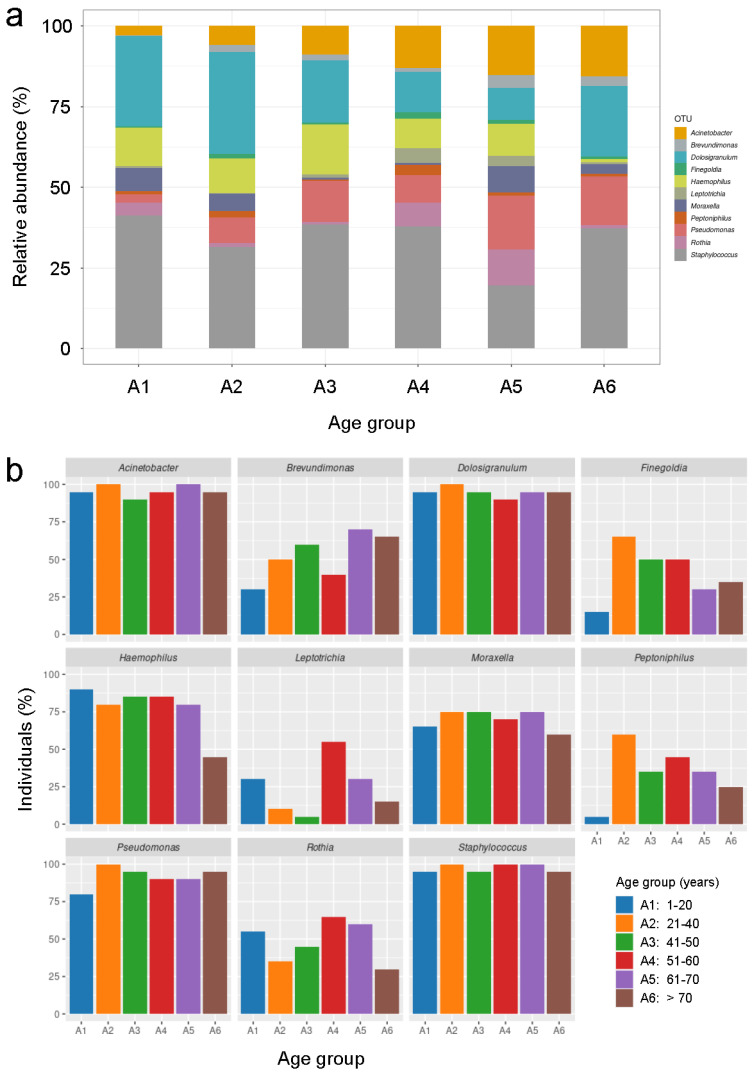
Taxonomic composition and frequency of appearance of the 11 bacterial genera which show significant differences between age groups. (**a**) Stacked bar charts showing the relative abundance (%) of the 11 bacterial genera indicated in the age groups established for this study. (**b**) Percentage of individuals, of the total included in this study, in which the indicated genera are present in the indicated age groups.

**Figure 5 ijms-24-04127-f005:**
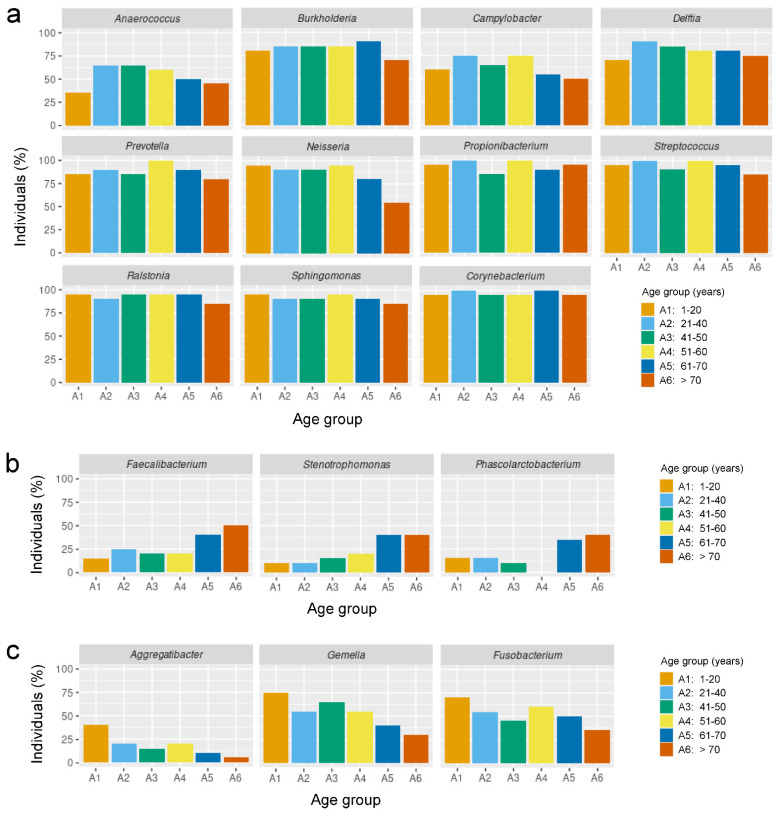
Frequency of appearance of potential biologically relevant bacterial genera. (**a**–**c**) Percentage of individuals, of the total included in this study, in which the indicated genera are present in the indicated age groups.

## Data Availability

Raw sequencing data of all 16S rRNA sequences, metadata, and abundance tables are available at the open access repository Figshare under the accession numbers 10.6084/m9.figshare.19785991 and 10.6084/m9.figshare.19786147, respectively.

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
