# Peer review of "Analysis of 16S rRNA Gene Sequence of Nasopharyngeal Exudate Reveals Changes in Key Microbial Communities Associated with Aging"

_ijms, 2023, doi:10.3390/ijms24044127_

Round 1

Reviewer 1 Report

This is a good written, diligent work.

Major concerns:

Title: Why mention the authors only SARS-CoV-2? What's about other respiratory viruses (e.g. influenza virus, RSV, metapneumonia virus, adenovirus)?

The method used enables detection of bacterial genus but not the species. There are important pathogenic bacteria to be detected; many people are carriers without falling ill (e.g. Streptococcus pneumoniae, Staphylococcus aureus, Chlamydia, Mycoplasma). I miss specific quantitative PCR to detect such pathogens. 16S rRNA gene sequence analysis enables only detection of genus but not species.

In section 4 (methods) I did not find a protocol to extract nucleic acids.

Minor concers:

Fig. 2a: Complete age groups or put into the figure legend.

Lines 211-213: The authors did not check the levels of sex hormones over age groups. To my opinion this is importent to assess microbiome diversity depending on sex and age.

Fig. 3a-f: Use same colours for each phylum.

Author Response

Reviewer 1.

Comments and Suggestions for Authors

This is a good written, diligent work.

We really thank these words.

Major concerns:

Title: Why mention the authors only SARS-CoV-2? What's about other respiratory viruses (e.g. influenza virus, RSV, metapneumonia virus, adenovirus)?

Thank you for this feedback. We initially mentioned SARS-CoV-2 because donors were sampled to diagnose a possible infection by that virus, but we agree with the reviewer’s point of view on this and the title has been amended in the revised version of the manuscript. Most people tested for SARS-CoV-2 infection when our samples were taken were uninfected subjects who simply had an infected relative or workmate and, as explained in the ‘Methods’ section (lines 521-527), all individuals with respiratory o any other kind of symptoms were excluded from our study, although they tested negative for SARS-CoV-2, to avoid any infections which potentially could alter the nasopharyngeal microbiota. In any case, the explanation of this issue has been improved in the revised manuscript.

The method used enables detection of bacterial genus but not the species. There are important pathogenic bacteria to be detected; many people are carriers without falling ill (e.g. Streptococcus pneumoniae, Staphylococcus aureus, Chlamydia, Mycoplasma). I miss specific quantitative PCR to detect such pathogens. 16S rRNA gene sequence analysis enables only detection of genus but not species.

We do not intend to focus on any particular species, nor on any group of bacteria that seem interesting according to our results (i.e. Corynebacterium genus, pathobiont bacteria, etc). That would be out of scope of our study, and  it should be the subject of future studies that want to delve into this issue.

In section 4 (methods) I did not find a protocol to extract nucleic acids.

In the ‘Methods’ section we mention that ‘The kit used for the PCR test was Novel Coronavirus (2019-nCoV) Real Time Multiplex RT-PCR kit (Detection for 3 Genes), manufactured by Shanghai ZJ Bio-Tech Co., Ltd. (Liferiver) and the CFX96 Touch Real-Time PCR Detection System (BioRad)’. That same kit was used for the extraction of nucleic acids, and this has now been clarified in the revised manuscript (lines 506-509).

Minor concers:

Fig. 2a: Complete age groups or put into the figure legend.

Thank you for this comment. As suggested, complete information about the age groups has been included in the figure 2 legend of the revised manuscript.

Lines 211-213: The authors did not check the levels of sex hormones over age groups. To my opinion this is importent to assess microbiome diversity depending on sex and age.

We agree on the relevance of sex hormone levels but, unfortunately, no serum samples from donors were available to test them. We would like to highlight that the samples were collected during the first wave of Covid-19 in Spain and hospital were overwhelmed.

Fig. 3a-f: Use same colours for each phylum.

We agree with this suggestion, and it has been amended in the revised manuscript.

Reviewer 2 Report

 The present manuscript have some fundamental problems; the first: The title Analysis of 16S rRNA gene sequence of nasopharyngeal exudate from SARS-CoV-2 negative individuals reveals changes in key microbial communities associated with aging, has no relation to the purpose of the study, it indicates that the cohort analyzed was not infected by the SARSCOV2 virus, there was no reason to mention the absence of this infection since it is a cohort of clinically healthy individuals, the entire design of the study is focused to the determination of the abundance and diversity of bacterial populations of the upper respiratory tract in clinically healthy individuals and its relationship with the age and gender of the individuals studied.

The entire manuscript from the introduction refers to viral infections, particularly the SARS-COV 2 virus, which is irrelevant for the study carried out since there is no comparison with a cohort of infected individuals.

The authors downplay their study from the very title, when the simple description of the population structure of this ecosystem could have been enough.

  On the other hand, the only differences found within the group studied are discreetly related to the gender of the individuals studied and they conclude that the differences could lie in the hormonal factor, although the number of samples studied for each of the 6 age and gender subgroups is of only 10 individuals.

  On the other hand, the characteristics of the different groups are not described. Inclusion, exclusion and elimination criteria were not considered in the design of the cohorts.

  My suggestion to the authors is to redesign the manuscript as a descriptive study of the bacterial population structure of the upper respiratory tract in a cohort of clinically healthy adult individuals. This would give more consistency to their study.

Author Response

Reviewer 2:

Comments and Suggestions for Authors

 The present manuscript have some fundamental problems; the first: The title Analysis of 16S rRNA gene sequence of nasopharyngeal exudate from SARS-CoV-2 negative individuals reveals changes in key microbial communities associated with aging, has no relation to the purpose of the study, it indicates that the cohort analyzed was not infected by the SARSCOV2 virus, there was no reason to mention the absence of this infection since it is a cohort of clinically healthy individuals, the entire design of the study is focused to the determination of the abundance and diversity of bacterial populations of the upper respiratory tract in clinically healthy individuals and its relationship with the age and gender of the individuals studied.

Please, see our response to reviewer 1´s concern about the title of this study. There, we explained why we initially proposed that title and how it has been amended now following the suggestions of reviewers 1 and 2.

The entire manuscript from the introduction refers to viral infections, particularly the SARS-COV 2 virus, which is irrelevant for the study carried out since there is no comparison with a cohort of infected individuals.

We agree with this comment. Thus, references to SARS-CoV-2 have been removed from the revised manuscript.

The authors downplay their study from the very title, when the simple description of the population structure of this ecosystem could have been enough.

This has been modified in the revised manuscript.

  On the other hand, the only differences found within the group studied are discreetly related to the gender of the individuals studied and they conclude that the differences could lie in the hormonal factor, although the number of samples studied for each of the 6 age and gender subgroups is of only 10 individuals.

The sex-associated differences found led us to speculate that they could be a consequence of the existence of differences in the sex hormones levels at the different ages, which is an amply demonstrated fact. Even through this is just speculation based on our results and on the previous knowledge, in the ‘Discussion’ section (lines 472-474) we clearly mention that the sample size of the sex groups is one of the limitations of our study.

  On the other hand, the characteristics of the different groups are not described. Inclusion, exclusion and elimination criteria were not considered in the design of the cohorts.

We honestly believe that the different groups and how patients were enrolled is clearly described at the beginning of the Methods (section 4.1), including the inclusion and exclusion criteria.

  My suggestion to the authors is to redesign the manuscript as a descriptive study of the bacterial population structure of the upper respiratory tract in a cohort of clinically healthy adult individuals. This would give more consistency to their study.

We really appreciate this comment as well. Reviewer 2´s suggestions have been taken into account to do several adjustments in the manuscript that, in our opinion, solve the issues mentioned by this reviewer.

Reviewer 3 Report

The authors have assessed the nasopharyngeal microbiome of Spanish healthy individuals. While the study is helpful in establishing a baseline for comparison of the healthy nasopharyngeal microbiota, the premise in correlation with COVID-19 infection seems flawed. Since they have not compared the healthy with infected patients, association with Sars-COV2 susceptibility in conclusions is baseless. The authors have used sweeping statements like ‘Our results provide a complete and valuable dataset that will be useful for future research aiming for studying the relationship between changes in the nasopharyngeal microbiome and susceptibility to or severity of multiple diseases, including COVID-19’. However, they have not described any dataset to support this statement. At best this study illustrates age dependent changes in selected bacterial communities of the nasopharyngeal microbiome. However, in this respect the limitations regarding socio-demographic perspectives of the dataset and limited number of samples of each gender in respective age group described, undermine its utility.

Author Response

Reviewer 3:

Comments and Suggestions for Authors

The authors have assessed the nasopharyngeal microbiome of Spanish healthy individuals. While the study is helpful in establishing a baseline for comparison of the healthy nasopharyngeal microbiota, the premise in correlation with COVID-19 infection seems flawed. Since they have not compared the healthy with infected patients, association with Sars-COV2 susceptibility in conclusions is baseless. The authors have used sweeping statements like ‘Our results provide a complete and valuable dataset that will be useful for future research aiming for studying the relationship between changes in the nasopharyngeal microbiome and susceptibility to or severity of multiple diseases, including COVID-19’. However, they have not described any dataset to support this statement. At best this study illustrates age dependent changes in selected bacterial communities of the nasopharyngeal microbiome. However, in this respect the limitations regarding socio-demographic perspectives of the dataset and limited number of samples of each gender in respective age group described, undermine its utility.

References to SARS-CoV-2 infection, including the one mentioned by reviewer 3, have been removed from the revised manuscript.

Round 2

Reviewer 1 Report

The authors addressed my concerns.

Author Response

We are glad to know that the concerns of this reviewer were satisfactorily addressed. 

Reviewer 2 Report

The revised version of the manuscript includes all my corrections and suggestions

Author Response

(The authors gave the same response as above.)

Reviewer 3 Report

The authors have tried to address the concerns raised against using the SARS-COV2 as the basis for their study design by removing most of the references to the infection. However, lack of any sociodemographic data as well as limited samples in each age group severely limit its utility as a baseline study to assess nasopharygeal microbiome. There is also the concern of use of a kit for nucleic acid extraction which is optimized for virus stabilization and RNA extraction for SARS-CoV2 detection. It is important to optimize collection and extraction procedures in microbial ecology to decrease bias introduced due to suboptimal extraction procedures. These may lead to underrepresentation of certain microbial communities. Furthermore there are ethical concerns regarding use of samples for purposes other than under the premise they were obtained.

Author Response

The authors have tried to address the concerns raised against using the SARS-COV2 as the basis for their study design by removing most of the references to the infection.

However, lack of any sociodemographic data as well as limited samples in each age group severely limit its utility as a baseline study to assess nasopharygeal microbiome.

We have indicated in our manuscript the limitations of our study (lines 455-465) that are like the ones found in other studies on nasopharyngeal microbiome (lines 466-482) and that we have detailed in our recent review (https://doi.org/10.1080/22221751.2023.2165970). We think, and the other 2 reviewers agree, that the study is of interest as the information about nasopharyngeal microbiome are rather limited.

There is also the concern of use of a kit for nucleic acid extraction which is optimized for virus stabilization and RNA extraction for SARS-CoV2 detection. It is important to optimize collection and extraction procedures in microbial ecology to decrease bias introduced due to suboptimal extraction procedures. These may lead to underrepresentation of certain microbial communities.

The extraction kit used was not indicated; just the kit used for PCR to determine if the individuals were or not positive for SARS-CoV-2. We are sorry for this mistake. It has now been indicated in the revised version (lines 492-494).

Furthermore there are ethical concerns regarding use of samples for purposes other than under the premise they were obtained.

Our study was approved by the Clinical Bioethics Committee of our Hospital with approval number 2020-10-12-HCUVA (lines 603-608). We have explained the reasons the exception to informed consent (lines 609-618); the most important one being that the samples were obtained during the first wave of the pandemic which was a life-threatening emergency with inadequate time to obtain consent. This is considered one of the exceptions to the informed consent (https://www.ncbi.nlm.nih.gov/books/NBK430827/).
